# Effects of cutaneous leishmaniasis on health-related quality of life: A longitudinal approach

Endi Lanza Galvão[1,2], Janaína de Pina Carvalho[2], Tália Santana Machado de Assis[2,3], Gláucia Cota[2], Sarah Nascimento Silva[2]*

**1** Programa de Pós-Graduação em Reabilitação e Desempenho Funcional, Departamento de Fisioterapia, Universidade Federal dos Vales do Jequitinhonha e Mucuri, Diamantina, Minas Gerais, Brasil, **2** Pesquisa Clínica e Políticas Públicas em Doenças Infecto-Parasitárias, Núcleo de Avaliação de Tecnologias em Saúde, Instituto René Rachou, Fundação Oswaldo Cruz, Belo Horizonte, Brasil, **3** Centro Federal de Educação Tecnológica de Minas Gerais, Contagem, Minas Gerais, Brasil

* sarah.nascimento@fiocruz.br

## Abstract

### Introduction

Cutaneous leishmaniasis (CL) is a neglected tropical disease that significantly affects patients' physical, psychological, and social well-being. Although previous studies have documented health-related quality of life (HRQoL) impairments associated with CL, most have relied on cross-sectional data, providing only a snapshot of the disease burden. Few studies have examined HRQoL progression over time or explored treatment effects from the patient's perspective.

### Objectives

This study aimed to evaluate changes in HRQoL among CL patients at three time points: before treatment, during treatment, and after treatment, using the Cutaneous Leishmaniasis Impact Questionnaire (CLIQ).

### Methods

A longitudinal observational study was conducted from October 2020 to April 2023 at a specialized center in Belo Horizonte, Brazil, involving 143 CL patients. The CLIQ was administered at diagnosis, during treatment (up to 30 days post-treatment), and at least 90 days after treatment initiation. Analyses included descriptive statistics, group comparisons using nonparametric methods, and correlation assessments based on rank-order relationships.

### Findings

Of the 143 patients enrolled, 120 completed all assessments. Median CLIQ scores improved significantly from pre-treatment (24, IQR 16–36) to post-treatment (15,

**Data availability statement:** All relevant data are within the manuscript and its Supporting Information files. Supporting InformationResearch data is avalable in: Silva, Sarah (2025), "Health-related quality of life in Brazilian patients with cutaneous leishmaniasis using EQ-5D_dataset", Mendeley Data, V1, doi: 10.17632/mh7zrrcxcy.1".

**Funding:** This study was funded by the following institutions: Programa Inova Fiocruz (https://portal.fiocruz.br), grant number VPPCB-008-FIO-18-2- 80. Programa Institucional de Bolsas de Iniciação Científica da Fundação Oswaldo Cruz (PIBIC-Fiocruz) (https://portal.fiocruz.br), which supported undergraduate student participation (no specific grant number). Fundação de Amparo à Pesquisa do Estado de Minas Gerais (FAPEMIG) (https://fapemig.br), which provided additional student research support (no specific grant number). Conselho Nacional de Desenvolvimento Científico e Tecnológico (CNPq) (https://www.gov.br/cnpq), grant number 302069/2022-4, awarded to GC (Gláucia Cota), as a research productivity fellowship. The funders had no role in study design, data collection and analysis, decision to publish, or preparation of the manuscript.

**Competing interests:** The authors have declared that no competing interests exist.

IQR 8.25–25; p<0.001). Notable improvements were observed in physical symptoms, emotional distress, and daily functioning. However, financial burden and social isolation persisted after treatment. Patient dissatisfaction with health services peaked during treatment but decreased afterward.

## Conclusion

Treatment significantly enhanced HRQoL in CL patients, particularly in the physical and functional domains. Nevertheless, persistent financial and emotional challenges underscore the need for comprehensive support interventions addressing economic and psychosocial dimensions, thereby promoting holistic care.

## Introduction

Cutaneous leishmaniasis (CL) is a neglected parasitic disease endemic to tropical and subtropical regions, affecting millions of people worldwide. This illness is characterized by debilitating cutaneous and mucosal lesions, which can substantially impair patients' quality of life [1,2], influencing physical [3], psychological [4], social, and economic dimensions [5,6]. Assessing health-related quality of life (HRQoL) is essential to fully understand the impact of CL, ensure that patient's well-being is considered, and guide both therapeutic interventions and public health policies.

Advances in medical treatments have expanded the range of therapeutic options available for diseases with long-term symptoms, such as CL, leading to increased interest in evaluating and comparing these interventions. In this context, assessing HRQoL is essential not only to evaluate the clinical effectiveness of treatments but also to capture their impact on patients' daily lives and overall well-being. Although studies on HRQoL provide valuable information on how diseases and treatments affect individuals' overall well-being, as far as we are concerned, the scientific literature on CL is still largely limited to cross-sectional studies [2,7–12]. These studies provide only one a static view of CL's effects on individuals, failing to capture changes over time. In this context, the dynamic nature of longitudinal studies allows for observing the evolution of health conditions and the impact of interventions on HRQoL, offering a more comprehensive and detailed understanding of quality of life. Identifying the areas most affected by the disease can facilitate the design of patient-centered treatments and management strategies, promoting interventions that address specific needs and improve the overall care experience [13].

Several tools have been proposed to assess HRQoL, from generic instruments such as the SF-36 (*Medical Outcomes Study 36-Item Form Health Survey*) [14] and the EQ-5D [15], developed by the EuroQol Group, to disease-specific tools such as the *Cutaneous Leishmaniasis Impact Questionnaire* (CLIQ) [16], designed to measure the impact of CL. However, the use of CLIQ still has limited trial experience regarding its potential to adapt to the characteristics of the disease, and further studies are needed to validate the tool's responsiveness [17].

In light of this, the present study aims to address this gap by comparing patients' HRQoL at three time points: prior to the disease onset, at diagnosis, during treatment (up to 30 days post-treatment), and after treatment completion, utilizing the CLIQ as the assessment tool. Furthermore, understanding the temporal dynamics of HRQoL in CL patients can guide the development of patient-centered care pathways and support the integration of psychosocial interventions into routine clinical management. These findings have potential implications for other endemic regions facing similar challenges, including limited access to care, social stigma, and treatment-related burdens, particularly in low- and middle-income countries where CL remains underprioritized in public health agendas.

## Methods

### Ethical considerations

This study was reviewed and approved by the Research Ethics Committee of the *René Rachou Institute*, *Oswaldo Cruz Foundation* (CAAE protocol # 28929220.0.0000.5091; approval number 3,918,626; March 16, 2020). Written informed consent was obtained from all participants. For individuals aged 12–17 years, both parental or guardian consent and the adolescent's assent were obtained, in accordance with Brazilian regulations and international ethical standards (Declaration of Helsinki; CIOMS, 2016).

### Study setting

The present study was carried out at the Leishmaniasis Reference Center of the *Rene Rachou Institute, Oswaldo Cruz Foundation*, in Belo Horizonte, Minas Gerais, Brazil.

### Study population and data collection

This longitudinal study was conducted between October 2020 and April 2023. Individuals aged 12 years or older with parasitological confirmation of active CL who agreed to participate in the study were included. The inclusion process was prospective, based on patients' access to healthcare services and the availability of the researchers involved. No exclusion criteria were applied. The sample size was not determined *a priori*; instead, all eligible patients presenting at the reference center during the study period who agreed to participate were included.

The validated version of the CLIQ [16] was administered to each patient by one of the four trained interviewers at the following timepoints: 1) before the start of the treatment, 2) during treatment (from the first dose to 30 days after the last dose), and 3) after treatment completion (at least 90 days after the first dose). To minimize missing data in the longitudinal analysis, two attempts were made to conduct interviews at time point 3, approximately 90–180 days (±14 days) after the start of treatment. When responses from both interviews at time point 3 were available, only data from the first interview were included in the analysis. This approach was adopted to ensure consistency and to prioritize the earliest measurement of post-treatment impact. Cases with missing follow-up data were excluded from the analysis. Sociodemographic information (sex, age, education) and clinical data—including clinical form of leishmaniasis [cutaneous, mucocutaneous (MCL), or mucosal leishmaniasis (ML)], presence of comorbidities, number of lesions, presence of ulceration or bacterial infection, case type (new or relapse) treatment used, and occurrence of adverse events—were recorded by the healthcare professional responsible for treatment and extracted from medical records using a structured form.

The CLIQ is a 25-item questionnaire covering two domains related to the health condition: overall impact of CL (questions 1–18) and perception of treatment and health services (questions 19–25). Scores for each item are summed to produce a total score ranging from 0 to 100. A higher CLIQ score indicates a greater impact of CL on the HRQoL. Questions 19 and 22, which address the participant's opinion of the medication used in treatment and the occurrence of adverse events, respectively, were not considered at the pre-treatment stage, as participants had not yet been exposed to the medication and therefore could not provide an informed opinion.

To assess individuals who reported some problem in the overall impact domain of CL, questionnaire variables were grouped into two categories: "no problems" and "some problems". Responses originally categorized by intensity (Nothing, Lightly, Neither a bit nor a lot, Moderately, Extremely) and frequency (Never, Almost never, Sometimes, Often, Frequently) were recoded, excluding only those indicating a total absence of problems (Nothing and Never). This allowed for a focused analysis of individuals who experienced some difficulty, providing detailed insights into the impacts of the disease on patients' HRQoL.

Although patients were asked about their impressions regarding treatment, the study did not aim to compare satisfaction across different drugs. Treatment selection was based on clinical manifestations, comorbidities, and medical judgment, while patient perspectives were assessed more broadly using the validated CLIQ instrument, which captures multiple dimensions of health-related quality of life.

### Data management and analysis

Data were entered into Microsoft Excel and analyzed using JASP 0.19.3.0. Descriptive statistics were used to characterize the study population, including measures of central tendency and dispersion for continuous variables, and frequencies and percentages for categorical variables. The Shapiro–Wilk test was applied to assess data normality. Because the data did not follow a normal distribution, nonparametric tests were selected for group comparisons. The Kruskal–Wallis test was employed to evaluate significant differences in the medians of CLIQ domain scores between the follow-up periods. When significant differences were detected, *post hoc* comparisons were performed using Dunn's test with Bonferroni correction to control for type I error.

Spearman's rank correlation coefficient was applied to examine the strength and direction of associations between selected continuous variables, with statistical significance set at $p < 0.05$. Missing data were managed using listwise deletion, whereby cases with incomplete data for the variables of interest were excluded from the corresponding analyses. For item-level responses to the questionnaire at the three treatment time points, only descriptive analyses were performed because the small sample size limited statistical power of the tests.

### Results

#### Participants' characteristics

A total of 143 patients diagnosed with CL, MCL, or ML were recruited, of whom 120 completed the longitudinal assessments, representing to a 16% loss to follow-up. Although the inclusion criteria permitted enrollment of individuals aged 12 years and older, no participants aged 12–14 years were enrolled during the study period; therefore, the youngest age included in the analysis was 15 years. At the baseline, the mean age of participants was 52 years (median 53; IQR 41–64), 73.4% were male (105/143), and 52.4% presented with a single lesion. The lower limbs were the most affected body region, accounting for 365 lesions (59.2%), followed by the mucous membranes of the nose and mouth, which comprised 112 lesions (18.1%). The distribution of clinical phenotypes was as follows: 74.1% CL (106/143), 11.1% ML (16/143), and 7.6% MCL (11/143). Baseline sociodemographic and clinical characteristics of participants who completed follow-up (n = 120) are presented in Table 1.

#### Participants' CLIQ scores

The overall median CLIQ scores at the three assessed time points were as follows: 1. prior to treatment, 24 (Q1: 16–Q3: 36); 2. during treatment, 23 (Q1: 16–Q3: 35.5); and 3. post-treatment, 15 (Q1: 8.25–Q3: 25). A significant positive linear correlation was observed across the scores at all time points, indicating consistency in the responses (time point 1 vs. time point 2, r = 0.762; time point 1 vs. time point 3, r = 0.498; time point 2 vs. time point 3, r = 0.635; p < 0.001). As shown in Fig 1, scores differed significantly between the pre-treatment and post-treatment time points (p < 0.001), indicating a greater disease impact at baseline. A significant difference was also observed between scores recorded during treatment

**Table 1. Sociodemographic and disease characteristics of participants (n = 120).**

| Variables | n (%) |
|---|---|
| **Gender** | |
| Male | 88 (73.4) |
| Female | 32 (26.6) |
| **Age** (years)* | |
| 15-29 y | 17 (14.2) |
| 30 a 59 y | 55 (45.8) |
| ≥60 y | 48 (40.0) |
| **Education level** | |
| Illiterate | 5 (0.04) |
| Middle or High School | 85 (70.8) |
| Higher education | 22 (18.3.) |
| NR | 8 (6.0) |
| **Clinical form of leishmaniasis** | |
| Localized | 90 (75.0) |
| Mucosal | 12 (10.0) |
| Mucocutaneous | 10 (8.3) |
| Disseminated | 8 (6.7) |
| **Associated comorbidities** | 71 (59.2) |
| **Number of lesions** * | |
| Only one | 62 (51.7.) |
| 2–6 | 35 (29.2) |
| 7 or more | 6 (5.0) |
| **Ulcerated Lesion** | 92 (76.7) |
| **Bacterial infection** | 15 (12.5) |
| **New case of leishmaniasis** | 105 (87.5) |
| **Treatment** | |
| Intravenous meglumine antimoniate | 43 (35.8) |
| Intralesional meglumine antimoniate | 34 (28.3) |
| Miltefosine | 31 (25.8) |
| Liposomal amphotericin B | 9 (7.5) |
| Fluconazole | 1 (1.0) |
| Intravenous meglumine antimoniate plus pentoxifylline | 2 (1.7) |
| **Adverse events** | 81 (67.5) |

*variable with missing information; the sum of events does not total 100%.

NR: not reported.

and those obtained after treatment completion (p < 0.001). In contrast, no significant difference was found between scores obtained before the start of treatment and those recorded during therapy.

## CLIQ domains

Domain scores were calculated by summing the individual items comprising each respective CLIQ domain. A significant improvement in patients' perception of the overall impact of CL was observed when comparing the pre-treatment and the post-treatment periods (p < 0.001), as well as between the treatment and post-treatment periods (p < 0.001). No significant difference was found between the pre-treatment and during-treatment time points (p = 0.058). In the domain evaluating

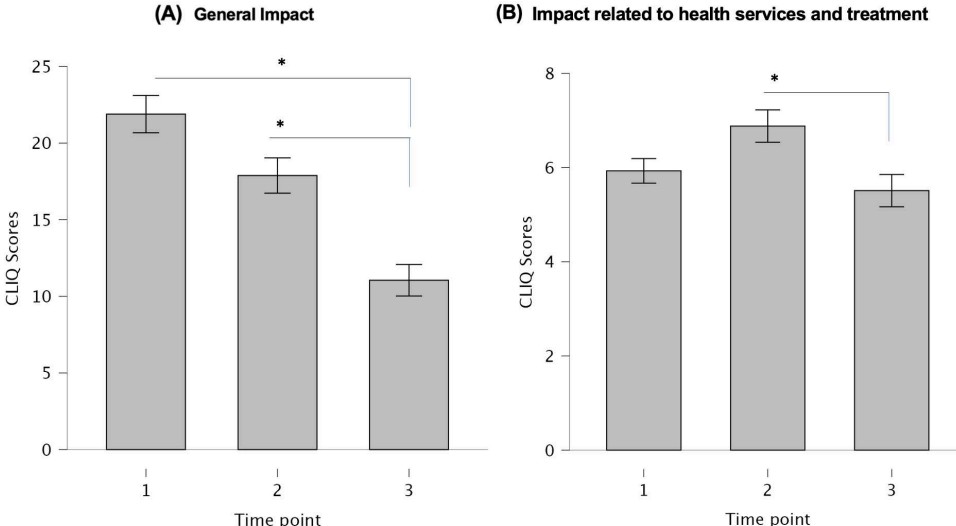

**Fig 1. Comparison of CLIQ overall scores over the follow-up period in patients with cutaneous leishmaniasis.** *Time points: 1. prior to treatment; 2. during treatment; and 3. post-treatment. *p < 0.05.*

patients' perceptions of treatment and access to health services, scores were significantly higher during treatment compared to the post-treatment period (p = 0.010). However, no significant differences were observed between the pre-treatment and during-treatment periods, nor between the pre-treatment and post-treatment periods for this domain. These findings are illustrated in Fig 2.

## CLIQ items

Table 2 presents the frequency of responses categorized as "some problems" among individuals reporting difficulties related to CL within the general health impact domain of the CLIQ, assessed before, during, and after treatment. Descriptive analysis revealed a decrease in perceived overall health impact (item 1), declining from 85.0% before treatment to 44.1% post-treatment. Difficulties related to physical activity (item 2) decreased from 51.6% to 20.8%, and challenges in work or study capacity (item 3) dropped from 50% to 22.5%. Reports of local symptoms (item 9) and concerns about skin

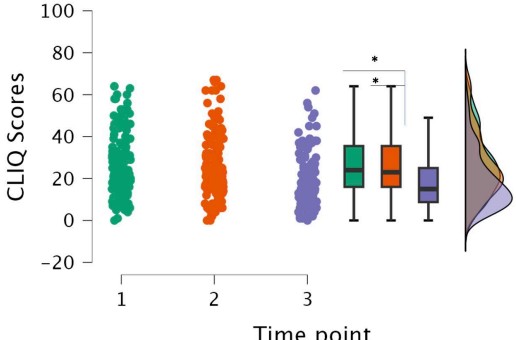

**Fig 2. Percentage of domain scores (A) General Impact (B) Perception about health services and treatment, and median/interquartile range (IQR) of CLIQ across follow-up periods.** *Time points: 1. prior to treatment; 2. during treatment; and 3. post-treatment. *p < 0.05.*

**Table 2. Frequency of 'Some Problems' responses in CLIQ Questionnaire's General Health Impact domain across three evaluation points: Before, During, and After Treatment (n = 120).**

| Item | Question | Number (%) reporting some problems | | | % change in numbers reporting some problems after treatment |
|------|----------|-----------------|----------------|----------------|------------------|
| | | Before the treatment | During the treatment | After treatment | |
| 1 | Has cutaneous leishmaniasis affected your overall well-being? | 102 (85.0) | 86 (71.6) | 53 (44.1) | − 55.8 |
| 2 | Has cutaneous leishmaniasis interfered with your physical activity? | 62 (51.6) | 52 (43.3) | 25 (20.8) | −30.8 |
| 3 | Has cutaneous leishmaniasis affected your ability to work (or study) in any way? | 60 (50.0) | 54 (45.0) | 27 (22.5) | − 27.5 |
| 4 | Has cutaneous leishmaniasis increased your health care expenses in any way? | 84 (70.0) | 88 (73.3) | 53 (44.1) | − 25.8 |
| 5 | Do you consider that cutaneous leishmaniasis has harmed your family financially? | 43 (35.8) | 50 (41.4) | 34 (28.3) | − 7.5 |
| 6 | Have you felt isolated from other people since you had cutaneous leishmaniasis? | 23 (19.2) | 23 (19.2) | 14 (11.6) | − 7.5 |
| 7 | Have you ever suffered from the feeling that you look different from people who do not have skin sores? | 56 (46.6) | 43 (35.8) | 35 (29.2) | − 17.5 |
| 8 | Have you ever had difficulty walking, changing clothes, or bathing because of your skin sore(s)? | 43 (35.8) | 34 (28.3) | 16 (13.3) | − 22.5 |
| 9 | Have you ever felt pain, burning, itching, or discomfort in the skin lesion areas? | 101 (84.1) | 88 (73.3) | 61 (50.8) | − 33.3 |
| 10 | Have you ever been nervous, sad, or scared because of cutaneous leishmaniasis? | 78 (31.6) | 65 (54.2) | 43 (35.8) | − 29.1 |
| 11 | Have you ever felt guilty or insecure because of cutaneous leishmaniasis? | 34 (28.3) | 30 (25.0) | 25 (20.8) | − 11.7 |
| 12 | Have you ever felt embarrassed because of your skin lesion(s)? | 47 (39.2) | 27 (22.5) | 35 (29.2) | − 10.0 |
| 13 | Have you ever missed work (or school) because of your skin lesion(s)? | 49 (40.8) | 48 (40.0) | 43 (35.8) | − 5.0 |
| 14 | Have you ever had difficulty having sex because of your skin lesion(s)? | 27 (22.5) | 27 (22.5) | 14 (11.7) | − 10.9 |
| 15 | How often have you had to pay someone to replace you in work or household activities to go to the health service? | 19 (15.8) | 20 (16.7) | 20 (16.7) | +0.8 |
| 16 | Have you had to change your dressing style because of other people's prejudice regarding your skin lesions? | 34 (28.3) | 41 (34.2) | 21 (17.5) | − 10.8 |
| 17 | How often have you avoided social activities with groups of people because of your skin lesion(s)? | 35 (29.2) | 35 (29.2) | 14 (11.7) | − 17.5 |
| 18 | How often do you depend on someone to accompany you to your cutaneous leishmaniasis treatment appointments? | 62 (51.7) | 63 (52.5) | 48 (40.0) | − 11.7 |

lesions (item 10) also diminished over the course of treatment. However, certain issues persisted to some extent, including the need to miss work (item 13), the requirement to pay for a substitute (item 15), perceived financial burden (item 5), and feelings of social isolation (item 6).

Regarding the perception of the medication used (item 19), 52.5% of participants rated it as "Very good" during treatment, increasing to 66.7% after treatment, representing an absolute increase of 14.2%. In contrast, item 22, which assesses the frequency of feeling unwell due to the medication, showed a 5.8% increase in the response "Sometimes" and a slight decrease of 1.7% in "Frequently". It is important to highlight that items 19 and 22 were not assessed at baseline, limiting direct comparisons with the initial condition. Additionally, reliance on health services for change wound dressing (item 23) decreased, with a 28.3% increase in the response "Never" post-treatment. Frequencies for all items related to treatment perception and health service use are presented in Table 3.

## Discussion

The results of the overall CLIQ scores indicate a significant positive impact of the various treatments on the patients' perceived quality of life, when comparing pre-treatment and post-treatment scores. As anticipated, no significant differences

**Table 3.** Frequency of responses of patients' perceptions of treatment and health services domain in CLIQ Questionnaire, across three evaluation points: Before, During, and After Treatment (n = 120).

| Item | Question | Number (%) reporting some problems | | | % change before and after treatment |
|------|----------|------------------|------------------|------------------|------------------|
| | | Before the treatment | During the treatment | After treatment | |
| 19 | What do you think about the medication you used to treat Cutaneous Leishmaniasis?* | | | | |
| | Very Good | 0 (0) | 63 (52.5) | 80 (66.7) | NA |
| | Good | 0 (0) | 47 (39.2) | 32 (26.7) | NA |
| | Middling | 0 (0) | 8 (6.7) | 4 (3.3) | NA |
| | Bad | 0 (0) | 1 (0.8) | 1 (0.8) | NA |
| | Very bad | 0 (0) | 1 (0.8) | 3 (2.5) | NA |
| 20 | What did you think about how you were welcomed by the health services when you were seeking diagnosis of Cutaneous Leishmaniasis? | | | | |
| | Very Good | 45 (44.6) | 56 (46.7) | 52 (43.3) | + 5.8 |
| | Good | 44 (36.7) | 41 (34.2) | 34 (28.3) | − 8.3 |
| | Middling | 18 (15.0) | 9 (7.5) | 16 (13.3) | − 1.7 |
| | Bad | 6 (5.0) | 7 (5.8) | 9 (7.5) | + 2.5 |
| | Very bad | 7 (5.8) | 6 (5.0) | 9 (7.5) | + 1.7 |
| 21 | What did you think about how you were welcomed by the health services when you were seeking treatment for Cutaneous Leishmaniasis? | | | | |
| | Very Good | 94 (78.3) | 87 (72.5) | 95 (79.2) | + 0.8 |
| | Good | 25 (20.8) | 26 (21.7) | 23 (19.2) | − 1.7 |
| | Middling | 1 (0.8) | 3 (2.5) | 1 (0.8) | 0 |
| | Bad | 0 (0) | 2 (1.7) | 1 (0.8) | − 0.8 |
| | Very bad | 0 (0) | 1 (0.8) | 0 (0) | 0 |
| 22 | How often have you felt sick because of the medications you took to treat Cutaneous Leishmaniasis?* | | | | |
| | Never | 0 (0) | 53 (44.2) | 56 (46.7) | NA |
| | Almost never | 0 (0) | 23 (19.2) | 13 (10.8) | NA |
| | Sometimes | 0 (0) | 30 (25.0) | 37 (30.8) | NA |
| | Often | 0 (0) | 6 (5.0) | 8 (6.7) | NA |
| | Frequently | 0 (0) | 8 (6.7) | 6 (5.0) | NA |
| 23 | How often have you relied on health services to provide you with supplies or to help change the wound dressing? | | | | |
| | Never | 74 (61.7) | 74 (61.7) | 96 (80.0) | + 28.3 |
| | Almost never | 7 (5.8) | 9 (7.5) | 5 (4.2) | − 1.7 |
| | Sometimes | 18 (15.0) | 14 (11.7) | 8 (6.7) | − 8.3 |
| | Often | 9 (7.5) | 12 (10.0) | 5 (4.2) | − 3.3 |
| | Frequently | 12 (10.0) | 11 (9.7) | 5 (4.2) | − 5.8 |
| 24 | How much do you care about the need to seek health services for the treatment of cutaneous leishmaniasis? | | | | |
| | Nothing | 65 (54.2) | 52 (43.3) | 74 (61.7) | + 7.5 |
| | Lightly | 11 (9.7) | 26 (21.7) | 13 (10.8) | + 1.7 |
| | Neither a bit nor a lot | 6 (5.0) | 5 (4.2) | 5 (4.2) | − 0.8 |
| | Moderately | 20 (16.7) | 23 (19.2) | 21 (17.5) | − 0.8 |
| | Extremely | 18 (15.0) | 14 (11.7) | 7 (5.8) | − 9.1 |

*(Continued)*

**Table 3.** (Continued)

| Item | Question | Number (%) reporting some problems | | | % change before and after treatment |
| --- | --- | --- | --- | --- | --- |
| | | Before the treatment | During the treatment | After treatment | |
| 25 | How long has it taken to get the tests done, medical appointments or hospitalizations related to Cutaneous Leishmaniasis? | | | | |
| | Nothing | 20 (16.7) | 29 (24.2) | 30 (25.0) | + 8.3 |
| | Lightly | 18 (15.0) | 17 (14.7) | 16 (13.3) | − 1.7 |
| | Neither a bit nor a lot | 15 (12.5) | 12 (10.0) | 13 (10.8) | − 1.7 |
| | Moderately | 32 (26.7) | 25 (20.8) | 33 (27.5) | − 0.8 |
| | Extremely | 35 (26.7) | 37 (30.8) | 28(23.3) | − 5.8 |

\* Items 19 and 22 were not administered at baseline, as they specifically assess the patient's perception of the treatment for Cutaneous Leishmaniasis. NA: not applicable.

were found between scores obtained before and during treatment, as symptom resolution of CL is typically perceived only after 90–180 days following treatment initiation [18].

The present findings suggest that although several areas of difficulty improved following treatment, economic and emotional challenges persisted, underscoring the need for additional support to address the financial and emotional dimensions of CL [6]. Analysis of items related to the impact of treatment and health services revealed increased variability in responses during and after treatment. This indicates that while some patients reported improvements and held positive perceptions of their care, others continued to experience difficulties or perceived the treatment process and health services negatively. This variability in responses suggests that patients' experiences with treatment were more heterogeneous, reflecting a broad spectrum of individual perceptions and outcomes. While treatment was effective for some, others faced distinct or additional challenges during the recovery process [19].

Although treatment resulted in noticeable improvements in patients' HRQoL and self-sufficiency, two key issues remained largely unchanged: financial burden and the feeling of social isolation. Despite a reduction in physical symptoms and increased independence, some patients continued to experience economic hardship due to treatment-related costs and loss of income, as well as feelings of social isolation, possibly linked to disease-related stigma or the inability to fully engage in social life. Although several studies have thoroughly examined the psychological impact of CL [1,4,20], the financial burden associated with the disease remains underexplored in the existing literature. Available evidence indicates that CL treatment can be economically burdensome, particularly in settings with limited access to quality health services and where travel and medication costs are substantial [5,21]. Recent evidence has further emphasized the substantial economic burden associated with CL treatment, underscoring significant out-of-pocket expenses and indirect costs related to productivity loss and transportation—factors that disproportionly affect vulnerable socioeconomic groups [6]. Accordingly, our findings highlight the need for complementary interventions, including financial assistance, psychological support programs, and patient and family support groups, to address the financial and emotional dimensions of CL that are often underestimated in clinical management, yet have a substantial impact on patients' quality of life.

This study found that perceived dissatisfaction with treatment and access to health services was greater during the treatment phase, suggesting that the challenges encountered at this stage may contribute to heightened emotional distress. These findings are consistent with previous research reporting elevated dissatisfaction among patients undergoing treatment, particularly in relation to access barriers and logistical difficulties [7]. Additionally, variations in responses between the treatment and post-treatment periods reflect a broader spectrum of experiences and perceptions, likely influenced by individual differences in emotional resilience and the availability of social support. These perceptions may also be shaped by factors such as injury severity, treatment response, and the extent of economic support received. This

 

echoes conclusions drawn in a qualitative study of Tunisian CL patients, who reported significant physical and emotional burden during treatment, along with frustration regarding limited efficacy and poor healthcare access, underscoring the importance of patient-centered care and the integration of psychosocial support into CL case management [19]. Although the present study was not designed to establish causal relationships, our findings offer valuable insights that can inform future research and clinical practice, contributing to a more comprehensive understanding of the challenges faced by patients.

Analysis of response frequencies for each questionnaire item revealed that the proportion of patients rating the medication as "very good" increased following completion of treatment, while ratings of "good," "fair," and "poor" decreased. However, a slight increase was observed in the number of patients who rated the treatment as "very poor". These evaluations may reflect a heightened perception of cure at the end of treatment. Furthermore, the use of parenteral medications may have contributed to less favorable assessments during the treatment period; however, with the achievement of cure, any transient discomfort may be disregarded. Additionally, an increase in the frequency of reported adverse events was observed throughout the treatment. These findings highlight the importance of continuous monitoring and adjustments during treatment to enhance patient satisfaction and manage adverse events effectively [22].

Hospitalization was not assessed as a study variable in this research. At the reference center, only patients treated with amphotericin B required inpatient care for drug administration, while all other treatments were delivered on an outpatient basis. Notably, only 7.5% of participants received amphotericin B. Although inpatient care was outside the scope of our analyses, future studies should consider evaluating hospitalization status and duration, as the need for inpatient treatment may impact patients' quality of life, treatment burden, adherence, and overall costs. Previous studies in CL have highlighted that economic costs, treatment logistics, and access to health services are important determinants of HRQoL [6,21], reinforcing the potential value of incorporating hospitalization into patient-reported outcome assessments.

The main limitations of this study include the lack of a representative sample of the broader population affected by CL, stemming from its single-center design and reliance on convenience sampling. Additionally, conducting interviews at the same health center where patients receive care may have introduced response bias—specifically, "gratitude bias"—whereby patients may have overstated improvements in their quality of life due to please the interviewer or to appear cooperative.

The longitudinal design may also have introduced biases due to loss to follow-up over time, complicating consistency in data collection. Moreover, the analytical approach adopted for item evaluation was predominantly descriptive and lacked methodological rigor, which may have limited the identification of definitive associations. Furthermore, the lack of control over external variables (such as variations in the treatment environment, adherence levels, social support, and comorbidities, among others), combined with the use of a non-representative sample, may have constrained the generalizability of the results.

Another limitation lies in the consolidation of intensity and frequency response categories in the CLIQ into a single category labeled "some problems". This simplification may hinder a comprehensive overview of patient improvements, potentially omit important information, and limit the detection of subtle yet clinically relevant changes. Furthermore, it fails to capture improvements that do not result in a shift to "no problems," such as a reduction from severe to moderate issues. This approach also diminishes the sensitivity and specificity of the assessment compared to more detailed versions of quality-of-life measurement instruments, thereby compromising the analysis of gradual and multidimensional improvements in patient health.

Despite these limitations, the study contributes meaningfully to the understanding of patients' lived experiences. While CL treatment significantly improves HRQoL in terms of physical symptoms and self-sufficiency, economic and emotional challenges persist as critical areas that requiring ongoing attention. The diversity in patient responses throughout the treatment process highlights the importance of personalized and integrated approaches that consider every dimension of patients' lives, rather than focusing solely on the physical symptoms of the disease.

This study supports the need for integrated public health responses to CL that go extend clinical cure, incorporating financial support mechanisms, psychosocial care, and community-based education aimed at reducing stigma and improving adherence. Such strategies could be embedded within national neglected tropical disease control programs, particularly in endemic regions where social vulnerability, geographic barriers, and resource constraints intensify the burden of illness. Tailored policies that recognize HRQoL as a critical outcome can foster more equitable and effective care for populations affected by CL.

## Acknowledgments

The authors would like to express their gratitude to Instituto René Rachou (Fiocruz Minas), Fundação Oswaldo Cruz, and Universidade Federal dos Vales do Jequitinhonha e Mucuri (UFVJM) for their invaluable support throughout the development of this research.

## Author contributions

**Conceptualization:** Endi Lanza Galvão, Gláucia Cota, Sarah Nascimento Silva.

**Data curation:** Endi Lanza Galvão, Janaína de Pina Carvalho, Sarah Nascimento Silva.

**Formal analysis:** Endi Lanza Galvão, Janaína de Pina Carvalho, Tália Santana Machado de Assis, Gláucia Cota, Sarah Nascimento Silva.

**Methodology:** Endi Lanza Galvão, Janaína de Pina Carvalho, Tália Santana Machado de Assis, Gláucia Cota, Sarah Nascimento Silva.

**Project administration:** Endi Lanza Galvão.

**Software:** Endi Lanza Galvão.

**Supervision:** Endi Lanza Galvão, Sarah Nascimento Silva.

**Writing – original draft:** Endi Lanza Galvão, Janaína de Pina Carvalho, Tália Santana Machado de Assis, Gláucia Cota, Sarah Nascimento Silva.

**Writing – review & editing:** Endi Lanza Galvão, Janaína de Pina Carvalho, Tália Santana Machado de Assis, Gláucia Cota, Sarah Nascimento Silva.

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
