## [Decision Letter · Decision Letter 0]

8 Sep 2025

PONE-D-25-35705
Effects of cutaneous leishmaniasis on the health-related quality of life: a longitudinal approach
PLOS ONE

Dear Dr. Silva,

Thank you for submitting your manuscript to PLOS ONE. After careful consideration, we feel that it has merit but does not fully meet PLOS ONE’s publication criteria as it currently stands. Therefore, we invite you to submit a revised version of the manuscript that addresses the points raised during the review process.

The manuscript was assessed by two independent reviewers that acknowledged the relevance of the study and the soundness of its methodological approach. Nevertheless, they made several comments and suggestions, mostly aimed at clarifying some aspects of the research method, that should be addressed by the authors. Although the reviewers also disclosed that they considered the overall report well written, both mention that there is still room for further language improvement. I suggest that the authors attempt to address in detail all points raised by the two reviewers, which I judged very constructive, and perform another round of language improvement in the revised version of the manuscript.

We look forward to receiving your revised manuscript.

Kind regards,

Albert Schriefer, M.D., Ph.D.

Section Editor

PLOS ONE

Journal Requirements:

“The authors would like to express their gratitude to Instituto René Rachou (Fiocruz Minas), Fundação Oswaldo Cruz, and Universidade Federal dos Vales do Jequitinhonha e Mucuri (UFVJM) for their invaluable support throughout the development of this research. We acknowledge funding from Programa Inova Fiocruz [Grant number VPPCB-008-FIO-18-2-80], which made this project possible. Additional support provided by Programa Institucional de Bolsas de Iniciação Científica da Fundação Oswaldo Cruz (PIBIC-Fiocruz) and Fundação de Amparo à Pesquisa do Estado de Minas Gerais (FAPEMIG) is gratefully acknowledged for enabling student participation. Gláucia Cota is a recipient of a research productivity grant from Conselho Nacional de Desenvolvimento Científico e Tecnológico (CNPq - grant number 302069/2022-4).”

“This study was funded by the following institutions:

Programa Inova Fiocruz (https://portal.fiocruz.br), grant number VPPCB-008-FIO-18-2-

80.

Programa Institucional de Bolsas de Iniciação Científica da Fundação Oswaldo Cruz

(PIBIC-Fiocruz) (https://portal.fiocruz.br), which supported undergraduate student

participation (no specific grant number).

Fundação de Amparo à Pesquisa do Estado de Minas Gerais (FAPEMIG)

(https://fapemig.br), which provided additional student research support (no specific

grant number).

Conselho Nacional de Desenvolvimento Científico e Tecnológico (CNPq)

(https://www.gov.br/cnpq), grant number 302069/2022-4, awarded to GC (Gláucia

Cota), as a research productivity fellowship.

The funders had no role in study design, data collection and analysis, decision to

publish, or preparation of the manuscript.”

Reviewers' comments:

Reviewer's Responses to Questions

**Comments to the Author**

1. Is the manuscript technically sound, and do the data support the conclusions?

Reviewer #1: Yes

Reviewer #2: Yes

2. Has the statistical analysis been performed appropriately and rigorously? 

Reviewer #1: N/A

Reviewer #2: Yes

3. Have the authors made all data underlying the findings in their manuscript fully available?

Reviewer #1: Yes

Reviewer #2: Yes

4. Is the manuscript presented in an intelligible fashion and written in standard English?

Reviewer #1: Yes

Reviewer #2: Yes

5. Review Comments to the Author

Reviewer #1: I think this is a well written article. However I have a few comments which needs to be addressed.

1. How were the patients diagnosed? Which according to them was the most satisfying and accurate?

2.Which according to the patients was the safest and best drug for treatment as far as satisfaction and permanent cure are concerned?

4.What was the definition of complete cure in these patients?

5. Please define the HIV status and other clinical parameters of these patients.

6. For how long were these patients hospitalised?

7. Were these patients followed up, if so for how long?

8. The english needs to be improved.

Reviewer #2: Overall Comment

This is a well-designed and clearly reported longitudinal study that fills an important gap in understanding the health-related quality of life (HRQoL) of cutaneous leishmaniasis patients. The results are valuable for informing patient-centered care and guiding public health interventions.

Additional Minor Comments

Abstract – Please include an introductory section to provide background on the study area and the rationale for the research.

Terminology – Ensure that Health-Related Quality of Life (HRQoL) is written in full before introducing the abbreviation in the abstract.

Ethical Considerations – The study population includes participants aged 12 years and above. Are 12-year-olds eligible to provide informed consent independently in Brazil? According to international ethical guidance (e.g., Declaration of Helsinki; CIOMS, 2016), children and adolescents should provide assent, while parental or guardian permission is required. Please clarify whether parental consent and child assent were obtained.

Methodology Consistency – In the results, it states: “A total of 143 patients diagnosed with cutaneous, mucocutaneous (MCL), or mucosal leishmaniasis.” Please ensure that the different clinical forms of leishmaniasis are consistently described in the methodology as well.

Proofreading – The manuscript would benefit from careful proofreading to improve sentence flow and clarity.

Sample Size – Please explain how the sample size was determined. Was a calculation performed or was it based on available cases?

Methodology-The inclusion criteria in the Methods section states that participants aged 12 years and older were eligible. However, in the Results section (Table 1), the youngest age category is reported as 15–29 years. This discrepancy needs clarification.

a) Were any participants aged 12–14 recruited?

b)If not, please explain why this group was included in the eligibility criteria but absent from the results.

c)If they were excluded post hoc, provide justification (e.g., ethical reasons, consent/assent procedures, or no eligible participants in this age band).

Loss to Follow-up – The reported 16% loss to follow-up is acceptable; however, presenting a brief comparison of baseline characteristics between participants who completed the study and those lost to follow-up would strengthen confidence in the findings. Were there notable demographic or clinical differences?

Socio-demographics Reporting – The manuscript reports 143 participants in the socio-demographics section, despite a 16% loss to follow-up (final n=120). Please clarify this discrepancy.

Discussion Citations – Several statements in the discussion would benefit from citations, particularly the finding that financial burden and social isolation persist despite treatment. Strengthening these points with references and linking them to policy implications would improve the discussion.

Discussion Line 287–288 – Please provide appropriate citations to support this statement.

6. PLOS authors have the option to publish the peer review history of their article (what does this mean?). If published, this will include your full peer review and any attached files.

Reviewer #1: No

Reviewer #2: No

---

## [Author Response · Author response to Decision Letter 1]

18 Sep 2025

Dear Editor,

We have made every effort to improve the text accordingly. The modifications implemented in this revised version are as follows:

- Add a letter to the reviews.

- Add a manuscript (R1).

Best Regards

---

## [Decision Letter · Decision Letter 1]

25 Sep 2025

Effects of cutaneous leishmaniasis on the health-related quality of life: a longitudinal approach

PONE-D-25-35705R1

Dear Dr. Silva,

We’re pleased to inform you that your manuscript has been judged scientifically suitable for publication and will be formally accepted for publication once it meets all outstanding technical requirements.

Kind regards,

Albert Schriefer, M.D., Ph.D.

Section Editor

PLOS ONE

Additional Editor Comments (optional):

Reviewer #2:

Reviewers' comments:

Reviewer's Responses to Questions

**Comments to the Author**

1. If the authors have adequately addressed your comments raised in a previous round of review and you feel that this manuscript is now acceptable for publication, you may indicate that here to bypass the “Comments to the Author” section, enter your conflict of interest statement in the “Confidential to Editor” section, and submit your "Accept" recommendation.

Reviewer #2: All comments have been addressed

2. Is the manuscript technically sound, and do the data support the conclusions?

Reviewer #2: Yes

3. Has the statistical analysis been performed appropriately and rigorously? 

Reviewer #2: Yes

4. Have the authors made all data underlying the findings in their manuscript fully available?

Reviewer #2: Yes

5. Is the manuscript presented in an intelligible fashion and written in standard English?

Reviewer #2: Yes

6. Review Comments to the Author

Reviewer #2: I have reviewed the resubmitted document and confirm that all comments have been satisfactorily addressed. I recommend the manuscript for publication.

7. PLOS authors have the option to publish the peer review history of their article (what does this mean?). If published, this will include your full peer review and any attached files.

Reviewer #2: No

---

## [Editor Report · Acceptance letter]

PONE-D-25-35705R1

PLOS ONE

Dear Dr. Silva,

I'm pleased to inform you that your manuscript has been deemed suitable for publication in PLOS ONE. Congratulations! Your manuscript is now being handed over to our production team.

Kind regards,

on behalf of

Dr. Albert Schriefer

Section Editor

PLOS ONE